# The lunar cycle drives migration of a nocturnal bird

**Gabriel Norevik**[ID]*, **Susanne Åkesson**[ID], **Arne Andersson, Johan Bäckman, Anders Hedenström**[ID]*

Centre for Animal Movement Research, Department of Biology, Lund University, Lund, Sweden

* gabriel.norevik@biol.lu.se (GN); anders.hedenstrom@biol.lu.se (AH)

**Data Availability Statement:** All relevant data are within the paper and its Supporting Information files.

**Funding:** The research received support from the Swedish Research Council (https://www.vr.se/) to

## Abstract

Every year, billions of seasonal migrants connect continents by transporting nutrients, energy, and pathogens between distant communities and ecosystems. For animals that power their movements by endogenous energy stores, the daily energy intake rates strongly influence the speed of migration. If access to food resources varies cyclically over the season, migrants sensitive to changes in daily energy intake rates may adjust timing of migration accordingly. As an effect, individuals adjusting to a common temporal cycle are expected to approach synchrony in foraging and movement. A large-scale periodic pattern, such as the dark–light cycle of the moon, could thus synchronize migrations across animal populations. However, such cyclic effects on the temporal regulation of migration has not been considered. Here, we show the temporal influence of the lunar cycle on the movement activity and migration tactics in a visual hunting nocturnal insectivore and long-distance migrant, the European nightjar, *Caprimulgus europeaus*. We found that the daily foraging activity more than doubled during moonlit nights, likely driven by an increase in light-dependent fuelling opportunities. This resulted in a clear cyclicity also in the intensity of migratory movements, with occasionally up to 100% of the birds migrating simultaneously following periods of full moon. We conclude that cyclic influences on migrants can act as an important regulator of the progression of individuals and synchronize pulses of migratory populations, with possible downstream effects on associated communities and ecosystems.

## Introduction

The approximate 30-day lunar cycle is a powerful periodicity globally affecting physiological and behavioral rhythms in a range of organisms [1], with documented effects on activity pattern and space use [2–5], predator–prey interactions [6–9], and reproduction [9–13]. By affecting foraging conditions in a predictable manner, the lunar cycle could potentially act as an important driver for the temporal progress of many animal migrants through its impact on daily energy intake rates during fuelling periods at stopovers [14]. However, studies on migration strategies generally assume static intake rates, while the temporal influence of periodically fluctuating fuelling conditions on migration speed has not been considered [14–16]. Within the nocturnal avian family of Caprimulgidae, several species are visual predators, whose

AH (621-2012-3585, 2016-03625) and SÅ (621-2013-4361, 2016-05342), and the Centre for Animal Movement Research (CAnMove), financed by a Linnaeus grant (349-2007-8690) from the Swedish Research Council (https://www.vr.se/) and Lund University (https://lunduniversity.lu.se/). The funders had no role in study design, data collection and analysis, decision to publish, or preparation of the manuscript.

**Competing interests:** The authors have declared that no competing interests exist.

**Abbreviations:** GLS, geolocation system; GPS, global positioning system; MDL, multisensor data logger.

activity patterns during breeding are influenced by the lunar cycle [9–11]. Dark nights restrict foraging activity to periods of twilight around dusk and dawn, while moonlit nights appear to relax this temporal constraint [9]. As migration speed is closely related to the time spent on fuelling at stopover and thereby the rate of fuelling [15], migratory Caprimulgids could minimize the overall time and energy spent on migration if their foraging effort results in high daily energy intake rates [17]. Hence, we expect that European nightjars adjust fuelling episodes during migration according to the lunar cycle in order to maximize daily foraging duration, and thereby reach maximum daily energy intake rates [14,17].

We studied how the lunar cycle affects the migration tactic of European nightjars migrating between north European breeding areas and wintering grounds in southern Africa [18] using archival data loggers (global positioning system [GPS; $n = 17$], geolocation system [GLS; $n = 12$], and multisensor data loggers [MDLs; $n = 10$]) to record the movements of 39 individuals (S1 Table). The GPS and GLS position data documented the location and duration of stationary periods within the Temperate and Sahel zones, respectively (Fig 1). We used wingbeat-induced acceleration data sampled by the MDL-tags to record daily flight activity patterns and evaluated the birds' flight activity relative to the lunar cycle using periodic regression [19].

## Results and discussion

The flight activity data revealed a predominantly sedentary lifestyle outside shorter episodes of migratory flights, with incidents of intermittent activity records around dusk and dawn, confirming the primarily crepuscular lifestyle of nightjars (Fig 2A, S1 Fig and S2 Data) [9–11]. We also found a periodic activity pattern during nighttime throughout the nonbreeding season, measured both as the daily number of activity registrations and daily number of hours with activity registrations, which were positively correlated with periodicity of the lunar cycle (linear mixed model: $\beta = 2.09$, SE = 0.230, $P < 0.001$, $R^2 = 0.285$, $var_{(Intercept)} = 0.078$, $SD_{(Intercept)} = 0.280$, $var_{(slope)} = 0.327$, $SD_{(slope)} = 0.572$, S3 Data). The longest daily activity episodes of the nightjars thus matched the most moonlit nights in the lunar cycle (Fig 2B), during which the daily duration of intermittent activity was almost 2.5 times longer than during dark nights without moonlight. The individual heterogeneity in the relationship between the lunar cycle and presumed daily foraging duration may be a result of individual variation in seasonal phenology of the tracked birds [18]. Because of the latitudinal effect on night length (Fig 2A), the full moon will have a weaker effect on daily foraging duration at higher latitudes than within Africa. As the birds' activity is recorded by the MDL-tag through its vertical axis, individual differences in tag orientation may also influence activity detection. Together these factors could contribute to individual heterogeneity in flight activity, as indicated by the variation in individual intercept.

Daily increase in foraging activity may be due to (i) an increase in time available for foraging [9] or (ii) reduced foraging efficiency if lunar-phobic prey becomes less active and therefore less available during moonlit nights [6]. To distinguish between these two hypotheses, we tested how the daily travel intensity among the tracked nightjars correlated with the lunar cycle. Under the prediction that migrants will commence movement as they reach their preferred departure fuel loads [15], we expect elevated daily travel intensities (i.e., a larger fraction of the population to undertake migratory flights) after periods of moonlit nights if moonlight promotes fuel accumulation. To quantify daily travel intensity, we extracted migratory nights (defined by ≥3 hours of continuous flight, as registered by the MDL-tags, or >100 km covered by the GPS-tracked birds, respectively) and calculated the daily fraction of tagged individuals performing migratory flights. Like the daily foraging activity, the seasonal distribution of migratory flight nights showed a clear periodic pattern, with reoccurring episodes when the

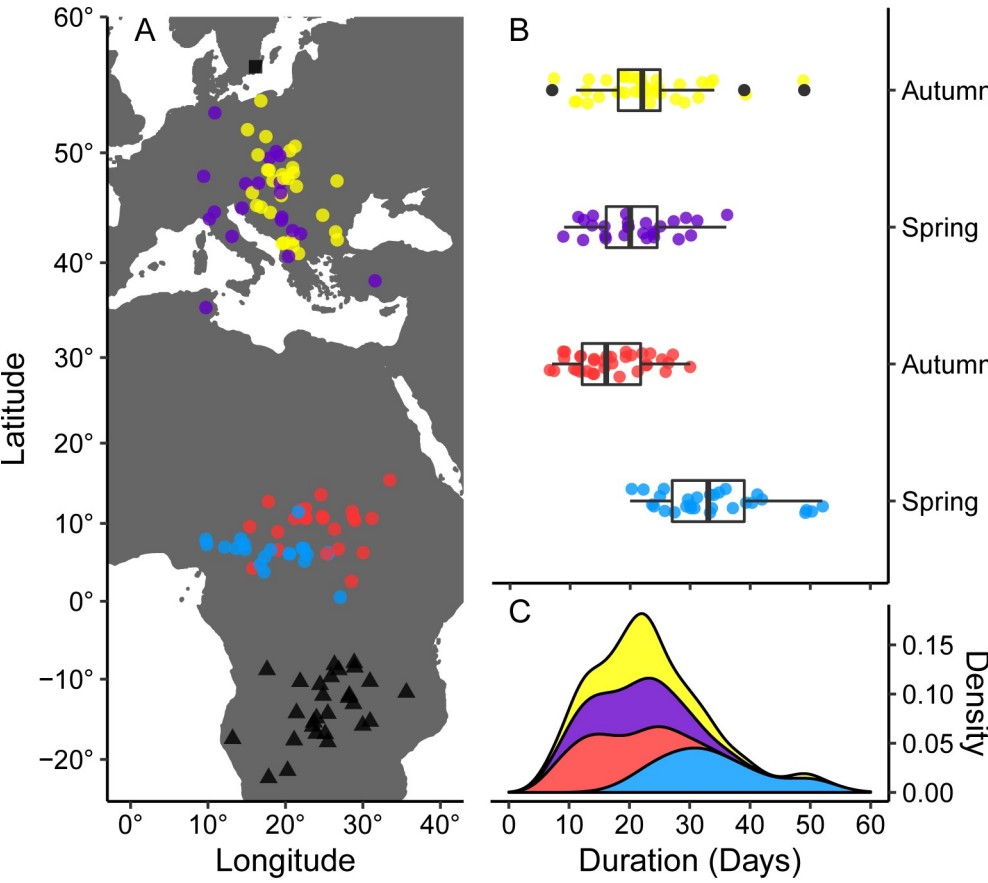

**Fig 1. Locations of stationary sites and stopover durations of migrating European nightjars.** (A) Locations of breeding (square), wintering (triangles), and stopover (circles) sites. Colors correspond to panels B and C. (B) Individual stopover durations in the Temperate and Sahel zones, respectively. Vertical bars indicate medians, boxes indicate 25th to 75th percentiles, whiskers indicate ranges, and black dots indicate outliers. (C) Pooled distribution of stationary durations showing a peak at 22 days. The map was plotted using the R package rworldmap [20]. Underlying data are found in S1 Data.

majority (occasionally 100%) of the tracked birds undertook migratory flights simultaneously (Fig 3A). When pooling the data relative to lunar cycle, the periodic pattern remained, with a peak in travel intensity 11 days following full moon (Fig 3B). This was also indicated by the significant relationship between lunar cycle and the tracked nightjar's flight nights (generalized linear mixed model: β = 0.618, SE = 0.086, $P < 0.001$, $var_{(Intercept)}$ = 0.021, $SD_{(Intercept)}$ = 0.145, $var_{(slope)}$ = 0.123, $SD_{(slope)}$ = 0.351, S5 Data), which is expected if moonlit nights enhance daily fuelling rates according to hypothesis (i) above. Although the overall pattern shows lunar cycle–associated cyclicity in migratory flights, there is a great deal of individual variation (Fig 3). Migratory flights do occur also around new moon, but they are more frequent after full moon (Fig 3B). Such variation may arise if arrival to a new stopover is mistimed in relation to the full moon in order to enjoy optimal foraging conditions. In turn, optimal departure on migratory flights may be affected by winds because a wind may change the utility of the current fuel load [21, 22], leading to knock-on effects at subsequent stopover and timing of departures.

The fuelling period is a time-consuming part of the migration, and variations in daily fuelling rates may have profound effects on the overall speed of migration [15]. Migrants whose objective is to minimize the overall time of migration are expected to depart from a stopover

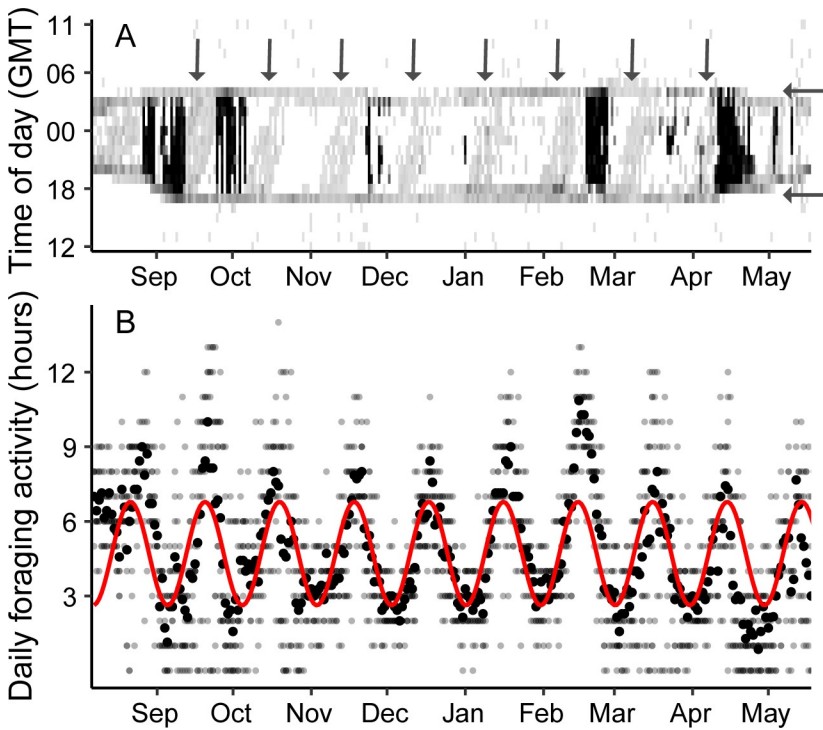

**Fig 2. Individual activity pattern and presumed daily foraging time during the nonbreeding season of European nightjars.** (A) Actogram showing the hourly activity index of an individual with no (white) to continuous (black) flight activity throughout the nonbreeding season. Gray corresponds to intermittent flight activity representing foraging flights. The bird's activity is confined to the night and only exceptionally is activity recorded during daylight hours, which demonstrates the species' crepuscular and nocturnal lifestyle. Black vertical lines represent periods of high activity spanning over several hours (indicative of migratory flights), and aggregations of black lines represent consecutive nights of migratory flights. Intermittent flight activity, presumably related to foraging, is concentrated towards dusk and dawn (horizontal arrows to the right). Following the daily cycle of the moon, intermittent flight activity is also registered during nighttime, resulting in diagonal bands throughout the nonbreeding season (vertical arrows). Notice the shorter activity periods from mid-April to May and in August, which reflect the shorter nights at northern latitudes. (B) The number of hours with registrations of presumed foraging activity per day and individual of nightjars in 2016–2017 (small dots) and daily means (large dots). Superimposed is a sine curve with a 29.54-day period, approximating the lunar cycle. Underlying data are found in S2 Data and S3 Data. GMT, Greenwich Mean Time.

when their marginal rate of gain in flight distance, i.e., the instantaneous speed of migration, drops to the expected migration speed [15,17]. Individuals that have reached the minimum fuel load required for the next flight step could thus depart as a response to reduced daily foraging duration (and associated fuelling rates) [14]. The most rapid decline in foraging duration occurs one week after full moon and provides a potential fixed proximate departure trigger that could result in the synchronous migration pattern of the birds (Fig 4, dashed line). Alternatively, if the migrant could time fuelling periods relative to the lunar cycle, it should strive to allocate stopovers to the most moonlit nights, i.e., by centering stopovers around full moon. By doing so, it would maximize the ratio of foraging time for a certain fuel load and total time spent at the stopover. This would result in a positive relation between stopover duration and departure timing, relative to the lunar cycle (Fig 4, solid line). However, when evaluating the stopovers in the annual cycle of the European nightjars (Fig 1B), we did not find any relationship between stopover duration and the lunar-related departure timing (S2 Table). This may be due to individuality in timing of the initiation of migration (onset of first fuelling period) not influenced by the lunar cycle and/or external factors such as knock-on effects from earlier annual events. For example, a tail wind may trigger departure from a stopover before the

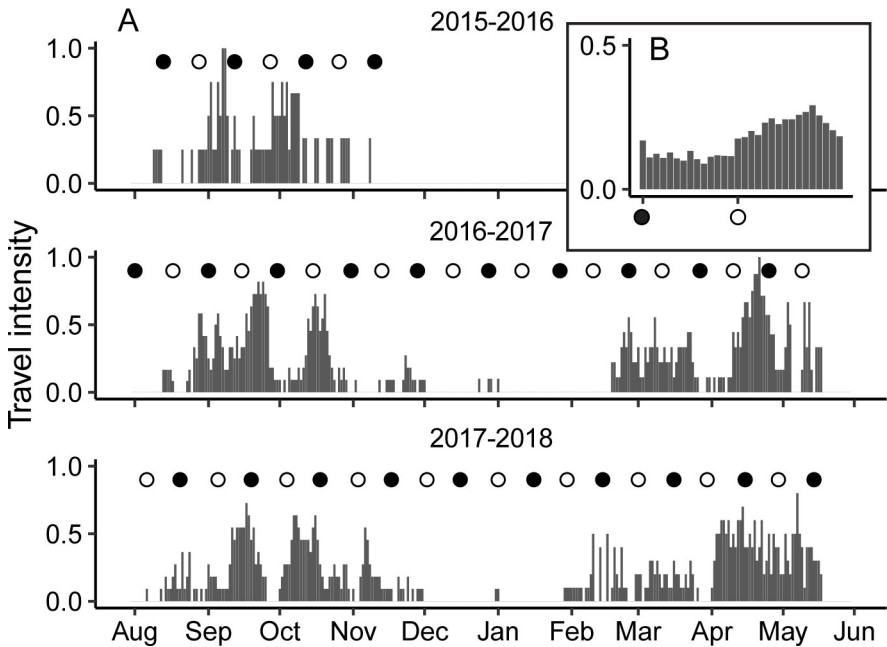

**Fig 3. Daily travel intensity of European nightjars.** (A) Daily fraction of individuals tracked during 2015–2018 undertaking a migratory flight, ranging from none (0) to all (1). Filled and open circles represent the timing of new and full moon, respectively. (B) Travel intensity data from the migration periods pooled relative to the lunar cycle. The daily fraction of traveling nightjars increases during the waning moon and peaks 11 days after the full moon. Underlying data are found in S4 Data.

optimal fuel load has been reached, which leads to suboptimal arrival with respect to the lunar cycle at the next stopover. Furthermore, the departure timings were consistently earlier than expected if the nightjars were striving to maximize the ratio between cumulative foraging time and total stopover duration (paired $t$ test: $t_{37} = -9.11$, $P < 0.001$, S3 Fig), suggesting that the birds respond to a fixed trigger relative to the lunar cycle when timing their departures. Our approach to use individual-based data loggers enabled us to record the nonbreeding movement of European nightjars and document the activity pattern related to the lunar cycle. However, we still lack information about feeding success and daily fuelling rates in relation to moon phases. It is therefore critical in future studies to focus on stopover ecology under the influence of the lunar cycle, preferably by studying individual departure decisions in relation to fuelling rates and fuel loads.

European nightjars clearly exhibit periodically fluctuating activity patterns on both daily and seasonal scales that are strongly associated with the lunar cycle and likely driven by light-dependent foraging opportunities [9]. This temporal arrangement of the annual schedule allows for substantially increased migration speeds through increased daily fuelling rates [5]. However, it is not clear how the timing of migration translates into other annual events and how nightjars are able to keep their circannual schedule while tracking the shifting cycles of the moon [23]. As an example, under climate change, mistiming of arrival to the breeding area may have long-lasting negative population and fitness consequences [24–25]. In particular, the influence of periodically varying foraging conditions on the progression of migration could aggravate cross-seasonal effects such as substantially delayed arrival to breeding grounds if the migrants need to adjust departure timing to a new cycle of peak conditions [26]. Interestingly, a few of the tracked nightjars spent surprisingly long stationary periods prior to crossing the Sahara Desert (Fig 1B), suggesting that they may have awaited a second lunar cycle before

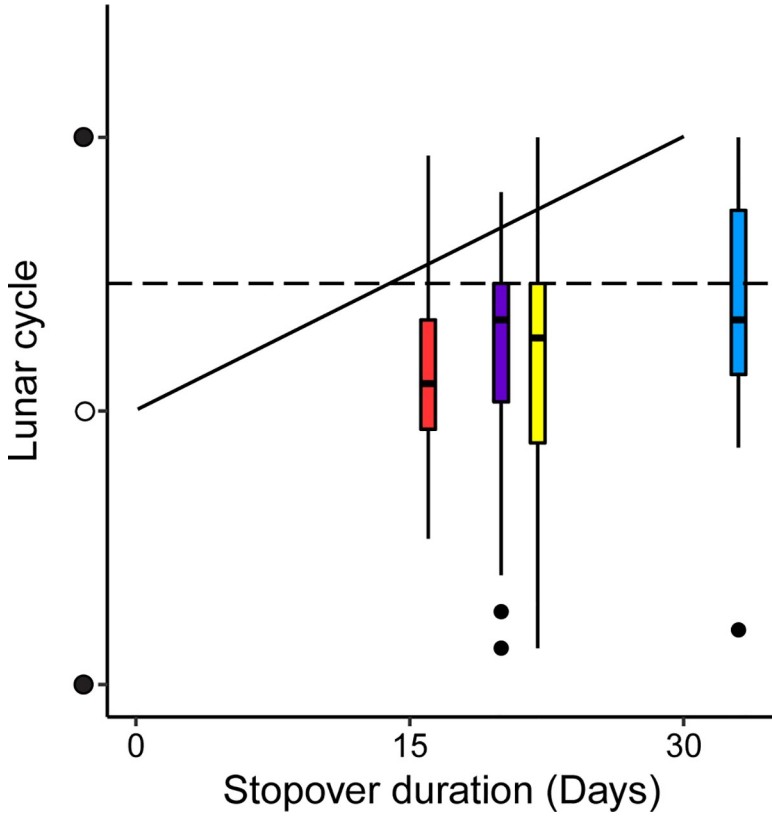

**Fig 4. Realized departure events and relative benefit of optimal departure timing under a range of stopover durations.** Box plots at median stopover durations corresponding to the four annual stopovers (see Fig 1). Bars indicate median departures, boxes indicate 25th to 75th percentiles, whiskers indicate ranges, and black dots show outliers. Departures could be triggered by dwindling fuelling rates as a fixed proximate factor regardless of stopover duration (horizontal dashed line). Alternatively, the nightjars may strive to maximize the ratio between cumulative foraging time and total stopover duration by centering the stopover around full moon. This would lead to a positive correlation between stopover duration and departure timing relative to the lunar cycle, increasing from full moon to new moon as stopover duration increases towards one lunar month (solid line). Underlying data are found in S1 Data.

commencing the barrier crossing. The link between arrival to the breeding ground and breeding success in nightjars is little known, although a temporal association between breeding events and the lunar cycle has been suggested for several species of Caprimulgids, including the European nightjar [10–11].

The lunar cycle is likely an important driver of the migration tactic of European nightjars, and this discovery may extend to other animal migrants whose foraging is affected by a periodically varying environment [1]. The lunar cycle has, for example, been directly linked to foraging behavior in nocturnal species of several groups that depend on vision for detecting and catching prey, such as seabirds [2,4,8], nightjars [10–11], and shorebirds, foraging in intertidal zones [27–28]. These groups comprise several species well known for their continent-wide migrations, whose timing of migration in relation to the lunar cycle remains unknown (except for the European nightjar as described here). Periodic fuelling conditions may affect both the timing of migration and its progression over large spatial scales, with potential downstream impact on both the migrants themselves and associated communities and ecosystems [29–31]. The degree to which large-scale movements of migratory populations are synchronized and temporally influenced could, for example, have potentially profound effects on local ecosystem functioning and pathogen dynamics [31–32]. Incorporating temporal dynamics of relevant

environmental factors, such as celestial bodies, could thus be a promising inroad to further our understanding of the seasonal pulses of animal migrations and their trophic effects.

## Materials and methods

### Ethics statement

The protocol follows the Swedish legislation for animal research (SJVFS 2019:9) and has been approved by the Malmö/Lund ethical committee for animal research (M33-13).

### Deployment and device settings

In total, 125 devices (65 archival GPS tags; NanoFix mini; Pathtrack, and 60 custom-made MDL-tags) were deployed on European nightjars in Sweden (16˚E, 57˚N) in 2015–2017. Trapped birds were equipped with either of the tag types using a full body harness as described in Norevik and colleagues [18]. Both logger types weigh less than 2.1 g, corresponding to <3% of the body mass of the tagged individual. The GPS tags were programmed to sample two fixes per night separated with 2 hours around local midnight during the periods 1 August–30 November and 1 March–30 May, respectively. During the rest of the season, when no migratory movements are expected [18], the tags were programmed to sample once every third night to save battery.

The MDL-tags were programmed to sample flight activity through acceleration, approximate altitude by air pressure, and geolocation by light. The acceleration was sampled in the z-axis, approximately parallel to gravity on a flying bird when mounted on the back, during 100 milliseconds at 100 Hz in the range ±4$g$. The mean of the values was subtracted from each of the 10 measurements to compensate for static gravity, and activity was considered as indicative of flight if at least 3 of the 10 values were greater than |g/3|. The sampling was repeated 10 times with 5-second intervals and the sampling procedure run every 5 minutes. For each run, the number of samples representing flight behavior was noted, i.e., (0, . . ., 10), where "0" indicates inactivity and "10" means that all samples suggested active flight. Every hour, a summary of results from all 12 runs were stored, representing the distribution of the samples between the different activity categories (0, . . ., 10), where (12, 0, 0, 0, 0, 0, 0, 0, 0, 0, 0) represents a bird with no recorded activity and (0, 0, 0, 0, 0, 0, 0, 0, 0, 0, 12) corresponds to a continuously flapping flying bird (see S4 Fig for an illustrative example and S2 Data). The pressure data were converted into altitude above sea level (m a.s.l.) using the hypsometric formula (International Organization for Standardization 1975: ISO 2533:1975):

$$z = \frac{T_0}{L}\left(\left(\frac{P_0}{P}\right)^{\frac{LR_0}{g}} - 1\right),$$

where $T_0$ is the temperature at sea level (assumed 288.15 K), $L$ is the altitudinal lapse rate of temperature (0.0065 degree K m$^{-1}$), $P_0$ is standard atmospheric pressure at sea level (1,013.25 hPa), $P$ is measured air pressure, $g$ is acceleration due to gravity (9.81 m s$^{-1}$), and $R_0$ is the universal gas constant (287.058 J kg$^{-1}$ K$^{-1}$).

Following a preprogrammed schedule, sequences of light-level measurements lasting for 5 days were distributed over the year, 20–24 August, 20–24 October, 20–24 December, 20–24 February, 20–24 April, and 20–24 May. The timing of measurement sequences was selected to avoid the equinox periods and to cover general stationary periods described in a previous geolocation study from the same site [18]. The geolocation data obtained from the MDL-tag mainly served as a control to provide approximate locations of the birds to be associated with key periods of the annual cycle. Measuring only selected periods substantially reduced the

amount of energy and memory required to process and store continuous light data. We also assumed that the European nightjars would remain within the longitudinal interval 20˚W to 50˚E [18], which longitudinally covers the African continent and corresponds to a local time interval of 4 hours and 40 minutes. This interval allows us to derive a threshold-based geolocation from the light data. For each measurement period, the light intensity was recorded every minute, and the maximum value during every 5-minute interval was stored.

Timing of transitions between night and day were extracted using a light threshold level of 2 (light range, 1–255) in the software IntiProc v 1.03 (Migrate Technology 2015), and a sun angle of −6˚ was selected for all devices by matching the derived positions with previously known stationary areas in Europe, the Sahel zone, and southern Africa for this population [18]. We allocated the data of the MDL-tags to three well-defined migratory episodes: movements within Europe, trans-Sahara migration, and the movements mainly covering sub-Saharan Africa [18]. Data on ambient light and air pressure were collected in 2 and 4 loggers, respectively. These data were used to estimate position and to corroborate registrations of migratory flights as recorded by the activity sensor (S4 Fig and S5 Fig).

Data from geolocators deployed during 2011–2014 were included in the analyses of the seasonal timing of migration involving the duration of time spent stationary within the Sahel and Temperate zones, respectively. See Norevik and colleagues [18] for handling of the light data.

## Data handling and statistics

We extracted flight segments from the activity data provided by the MDL-tags using two steps (S4 Fig). First, we assigned hours with high activity level (seven or more registrations of activity level 6 or above) as core periods of flight. This usually resulted in periods of several hours with high activity. Secondly, we added the number of 5-minute registrations of activity values of 7 or above in the hour immediately before (pre-core) and after (post-core) the core period to the total duration of flight. Each flight segment thus included a core period of at least one hour, with high activity values and the number of 5-minute registrations added in each end of the core period registered during the pre- and post-core hours, respectively. We noted that the hourly altitude estimate showed rapid increase also after pre-core hours, with only singles of 5-minute registrations of high activity (S5 Fig). We therefore treated these registrations as a part of the flight segment and included them when quantifying the flight durations.

As European nightjars are known to mainly be motionless in daytime and restrict their activity to foraging flights at dusk and dawn, except during periods of moonlit nights, all recorded activity outside the flight segments was defined as potential foraging activity [10,33–34]. Obviously, we cannot differentiate foraging from flights driven by other motivations, such as commuting or local movements, but based on the limited amount of movements observed in Caprimulgids [9,35–36], a major fraction of the activity outside segments of migratory flights is likely related to foraging. We evaluated the nightjars' activity relative to the moon cycle using periodic regression and linear mixed models ("lme4" package in R version 3.4.3. [37]). The 29.54-day moon cycle was divided into 360˚ (or $2\pi$ radians) to give each lunar day an angular equivalent, $\theta$ (theta). We set individual as random intercept to allow between-individual variation in activity and included $\theta$ as random slope to account for temporal heterogeneity in activity relative to the lunar cycle. Through an iterative process, we shifted the sinusoidal curve one day at a time to find the date with the best fit, corresponding to the temporal relationship between moon cycle and the birds' activity pattern. The distribution of individual flight versus non-flight nights relative to the moon cycle was investigated using a generalized linear mixed model with a binomial distribution and individual and $\theta$ as random intercept and slope, respectively.

## Supporting information

**S1 Data. Individual stopover duration and departure timing.** Contains data on duration, departure timing, and expected departure timing relative to the lunar cycle per individual, region, and season. Departure timing and expected departure timing are the number of days from new moon. Expected departure timing was derived under the precondition that individuals center their stopovers around full moon to maximize available foraging time per stopover day.
(XLSX)

**S2 Data. Raw activity data.** Contains 13 columns, "Timestamp," "Acc[0]" to "Acc[10]," and "Id." "Acc[0]" to "Acc[10]" correspond the number of sampling events per 5-minute run, and each row contains distributions of 12 runs during the preceding hour. See "Deployment and device settings" in the Materials and methods section for a detailed description of the sampling procedure and S4 Fig for an illustrative example.
(XLSX)

**S3 Data. Number of hours with recorded foraging activity per day and individual.** Contains four columns, "Date," "Id," "N hours," and "Theta." "N hours" is the number of hours per day with recorded foraging activity per individual, and "Theta" is the angular equivalent for each day in the lunar cycle. See "Data handling and statistics" in the Materials and methods section for how foraging activity was assessed and quantified.
(XLSX)

**S4 Data. Daily fractions of migrating nightjars.** Contains five columns, Date, Flight (n ind.), Total (n ind.), Flight fraction, and Theta. "Flight" denotes the number of individuals undertaking migratory flights, and "Total" corresponds to the number of individuals tracked per date. "Flight fraction" is the ratio between "Flight" and "Total," and "Theta" is the angular equivalent for each day in the lunar cycle. See "Data handling and statistics" in the Materials and methods section for how "Theta" was derived.
(XLSX)

**S5 Data. Recorded flight nights per individual.** Contains three columns, Date, Individual, and Flight. Flight denotes if the individual undertook a migratory flight (1) or not (0). See "Data handling and statistics" in the Materials and methods section regarding the assessment of migratory flights.
(XLSX)

**S1 Fig. Actogram of individual activity derived from MDL.** Actogram showing the hourly activity index of all individuals with no (white) to continuous (black) flight activity throughout the nonbreeding season. Gray corresponds to intermittent flight activity representing foraging flights. The birds' activity is confined to the night and only exceptionally is activity recorded during daylight hours, which demonstrates the species' crepuscular and nocturnal lifestyle. Black vertical lines represent periods of high activity spanning several hours (indicative of migratory flights), and aggregations of black lines represent consecutive nights of migratory flights. Intermittent flight activity, presumably related to foraging, is concentrated towards dusk and dawn. Following the daily cycle of the moon, intermittent flight activity is also registered during nighttime, resulting in diagonal bands throughout the nonbreeding season. Notice the shorter activity periods from mid-April to May and in August, which reflect the shorter nights at northern latitudes. Individual activity data were recorded from 15 July (or from the date of mounting; individuals XD86 and XD87) to the date of recovery of the tag (or when the tag stopped; individuals X500, X523, X526, and X531). Underlying data are found in

S2 Data. MDL, multisensor data logger.
(TIF)

**S2 Fig. Realized departure events and relative benefit of optimal departure timing under varying stopover durations.** Bars indicate median departures, boxes indicate 25th to 75th percentiles, whiskers indicate ranges, and black dots indicate outliers. Lines represent the relative difference in cumulative foraging time under median stopover durations for each region (Temperate and Sahel zones) and season (autumn and spring), respectively. Filled and open circles on the x-axis represent the timing of new and full moon, respectively. The effect of optimal timing relative to the lunar cycle is expected to vary depending on the duration of the stopover. With the estimated increase in daily foraging duration during moonlit nights, an optimal timing of an autumn stop in the Sahel zone with a median duration of 17 days would increase the cumulative foraging time 22% relative to average. In contrast, for a spring stop with a median duration of 33 days, the corresponding gain is only about 4%. For the median stopover durations in the Temperate zone of 22 and 25 days for spring and autumn, respectively, optimal timing would result in intermediate gains. As stopover duration increases, the optimal departure timing relative to the lunar cycle will shift. Contrary to expectations, median departure timing was rather similar between sites occurring 2–8 days after full moon. Underlying data are found in S1 Data.
(TIF)

**S3 Fig. Observed versus expected timing of departures in relation to lunar cycle.** Observed versus expected timing of departures in relation to lunar cycle (filled circles = new moon; open circles = full moon). Solid line represents "Observed" = "Expected" and colors represent stopover zone (T = Temperate; S = Sahel) and season, as in the main text. Overall, birds departed earlier than expected if they would strive to maximize the ratio between foraging time and stopover duration. Filled and open circles on the axes represent the timing of new and full moon, respectively. Underlying data are found in S1 Data.
(TIF)

**S4 Fig. Example of hourly activity and altitude measurements.** Hourly measurement of activity (bottom) and barometric-based altitudes (top) of European nightjar as sampled by a MDL-tag during a period of 2 days, starting and ending at noon. Circle size corresponds to the number of registrations observed per hour (1–12). An extended period of flight activity is initiated during hour 20 the second night (*t0*), resulting in a relatively large and rapid increase in altitude. At hour 01 (*t1*), the first low activity registration occurs, and the altitude has dropped and continues to be relatively stable throughout the rest of the night. This translates into a core period of 4 hours (21–00). As there are 11 registrations of high activity levels and only one registration of a low activity level both in the preceding hour (20) and the following hour (01), 55 minutes are added in both ends of the core period, resulting in an approximated flight segment of 5 hours and 50 minutes. Note that in both the first and the second night, there are several registrations of intermediate activity levels in the evening and morning, which likely are associated with foraging events. Also note how the altitude estimate gradually changes over the day as the sampled barometric pressure varies, likely due to, e.g., the passage of a low-pressure system. Underlying data are found in S2 Data. MDL, multisensor data logger.
(TIF)

**S5 Fig. Altitude change relative to activity registrations during migration initiation.** The barometric derived altitude measurement is taken hourly after the last of a series of 12 activity samplings. Each activity sampling consists of ten 5-minute registrations, and the number of positive activity registrations assigns it to one of eleven groups, 0–10; i.e., if there are no positive registrations in the activity sample, it is assigned to bin "0"; if there were three positive

registrations, it is assigned to bin "3." After one hour, all 12 activity samplings are distributed between the bins, and the activity pattern is stored. The change in altitude is the difference between the measurements taken during the pre-core hour and the hour before. Digits on the x-axis represent the number of 5-minute registrations during the pre-core hour, with values of 7 or higher indicating high activity. As the number of registrations increases, the change in altitude increases, indicating that high activity registrations at the initiation of a flight segment are part of the migratory flight. Underlying data are found in S2 Data.
(TIF)

**S1 Table. Devices and sampling periods.** Start and stop date for each device used. Tag type refers to GLS, geolocation system GPS, global positioning system; MDL, multisensor data logger.
(DOCX)

**S2 Table. Model output.** Linear mixed model with departure timing relative to lunar cycle as the dependent variable and the interaction between stopover duration and zone/season as the independent variable. Individual was set as random intercept ($var = 14.26$, $SD = 3.788$) and $\theta$ as random slope ($var = 0.004$, $SD = 0.059$).
(DOCX)

## Acknowledgments

We are grateful to Urban Rundström for his help during fieldwork. This is a report from CAnMove (Centre for Animal Movement Research) at Lund University.

## Author Contributions

**Conceptualization:** Gabriel Norevik, Susanne Åkesson, Anders Hedenström.

**Data curation:** Gabriel Norevik.

**Formal analysis:** Gabriel Norevik.

**Funding acquisition:** Susanne Åkesson, Anders Hedenström.

**Investigation:** Gabriel Norevik.

**Methodology:** Gabriel Norevik, Arne Andersson, Johan Bäckman.

**Project administration:** Anders Hedenström.

**Resources:** Anders Hedenström.

**Software:** Arne Andersson, Johan Bäckman.

**Supervision:** Susanne Åkesson, Anders Hedenström.

**Validation:** Arne Andersson.

**Visualization:** Gabriel Norevik.

**Writing – original draft:** Gabriel Norevik.

**Writing – review & editing:** Gabriel Norevik, Susanne Åkesson, Arne Andersson, Johan Bäckman, Anders Hedenström.

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
