## [Editor Report · Decision Letter 0]

4 Jul 2019

Dear Dr Norevik, 

Thank you for submitting your manuscript entitled "The lunar cycle drives migration of a nocturnal bird" for consideration as a Research Article by PLOS Biology.

Your manuscript has now been evaluated by the PLOS Biology editorial staff, as well as by an academic editor with relevant expertise, and I'm writing to let you know that we would like to send your submission out for external peer review. HOWEVER, I note that you have submitted it as a full Research Article. My previous discussions with Anders were on the basis that this should be considered as a Short Report, and the Academic Editor agrees that this would be the most appropriate format.

IMPORTANT: In order to switch the format you'll need to do two things - a) change the article type from Research Article to Short Reports when completing the submission (see below), and b) because the Short Report format only allows 3-4 Figures, and your submission currently has 6 main Figs, you'll need to either combine some of them into multi-panel Figs or move some to the Supplement, and adjust the text accordingly. 

IN ADDITION, before we can send your manuscript to reviewers, we need you to complete your submission by providing the metadata that is required for full assessment. To this end, please login to Editorial Manager where you will find the paper in the 'Submissions Needing Revisions' folder on your homepage. Please click 'Revise Submission' from the Action Links and complete all additional questions in the submission questionnaire.

**Important**: Please also see below for further information regarding completing the MDAR reporting checklist. The checklist can be accessed here: https://plos.io/MDARChecklist

Please re-submit your manuscript and the checklist, within two working days, i.e. by Jul 08 2019 11:59PM.

Kind regards,

Roli Roberts

Senior Editor

PLOS Biology

INFORMATION REGARDING THE REPORTING CHECKLIST:

PLOS Biology is pleased to support the "minimum reporting standards in the life sciences" initiative (https://osf.io/preprints/metaarxiv/9sm4x/). This effort brings together a number of leading journals and reproducibility experts to develop minimum expectations for reporting information about Materials (including data and code), Design, Analysis and Reporting (MDAR) in published papers. We believe broad alignment on these standards will be to the benefit of authors, reviewers, journals and the wider research community and will help drive better practise in publishing reproducible research. 

We are therefore participating in a community pilot involving a small number of life science journals to test the MDAR checklist. The checklist is intended to help authors, reviewers and editors adopt and implement the minimum reporting framework. 

IMPORTANT: We have chosen your manuscript to participate in this trial. The relevant documents can be located here:

MDAR reporting checklist (to be filled in by you): https://plos.io/MDARChecklist

**We strongly encourage you to complete the MDAR reporting checklist and return it to us with your full submission, as described above. We would also be very grateful if you could complete this author survey:

https://forms.gle/seEgCrDtM6GLKFGQA

Additional background information:

Interpreting the MDAR Framework: https://plos.io/MDARFramework

Please note that your completed checklist and survey will be shared with the minimum reporting standards working group. However, the working group will not be provided with access to the manuscript or any other confidential information including author identities, manuscript titles or abstracts. Feedback from this process will be used to consider next steps, which might include revisions to the content of the checklist. Data and materials from this initial trial will be publicly shared in September 2019. Data will only be provided in aggregate form and will not be parsed by individual article or by journal, so as to respect the confidentiality of responses. 

Please treat the checklist and elaboration as confidential as public release is planned for September 2019.

We would be grateful for any feedback you may have.

---

## [Decision Letter · Decision Letter 1]

12 Aug 2019

Dear Dr Norevik,

Thank you very much for submitting your manuscript "The lunar cycle drives migration of a nocturnal bird" for consideration as a Short Report by PLOS Biology. As with all papers reviewed by the journal, yours was evaluated by the PLOS Biology editors as well as by an Academic Editor with relevant expertise and by three independent reviewers. Based on the reviews, we will probably accept this manuscript for publication, assuming that you modify the manuscript to address the concerns raised by the reviewers.

IMPORTANT: Please attend to the following:

a) Address the concerns raised by the reviewers, including the request by reviewer #3 that you present some analysis of heterogeneity between individuals.

b) Address my Data Policy request below, ensuring that you supply the data requested and indicate clearly in each Figure legend where the data can be found.

We expect to receive your revised manuscript within two weeks. Your revisions should address the specific points made by each reviewer. In addition to the remaining revisions and before we will be able to formally accept your manuscript and consider it "in press", we also need to ensure that your article conforms to our guidelines. A member of our team will be in touch shortly with a set of requests. As we can't proceed until these requirements are met, your swift response will help prevent delays to publication.

Please note that you may have the opportunity to make the peer review history publicly available. The record will include editor decision letters (with reviews) and your responses to reviewer comments. If eligible, we will contact you to opt in or out.

Early Version: Please note that an uncorrected proof of your manuscript will be published online ahead of the final version, unless you opted out when submitting your manuscript. If, for any reason, you do not want an earlier version of your manuscript published online, uncheck the box. Should you, your institution's press office or the journal office choose to press release your paper, you will automatically be opted out of early publication. We ask that you notify us as soon as possible if you or your institution is planning to press release the article.

Sincerely,

Roli Roberts

Senior Editor

PLOS Biology

ETHICS STATEMENT:

The Ethics Statements in the submission form and Methods section of your manuscript should match verbatim. Please ensure that any changes are made to both versions.

-- Please include the full name of the IACUC/ethics committee that reviewed and approved the animal care and use protocol/permit/project license. Please also include an approval number if one was obtained.

-- Please include the specific national or international regulations/guidelines to which your animal care and use protocol adhered. Please note that institutional or accreditation organization guidelines (such as AAALAC) do not meet this requirement.

-- Please include information about the form of consent (written/oral) given for research involving human participants. All research involving human participants must have been approved by the authors' Institutional Review Board (IRB) or an equivalent committee, and all clinical investigation must have been conducted according to the principles expressed in the Declaration of Helsinki.

DATA POLICY:

We note that you say that the raw flight activity data have been deposited in Dryad; many thanks for doing this. However, we can't currently access the data, and will need to check it - please could you send login details or a reviewer link? In addition, we ask that all individual quantitative observations that underlie the data summarized in the figures and results of your paper be made available in one of the following forms:

Regardless of the method selected, please ensure that you provide the individual numerical values that underlie the summary data displayed in Figs 1, 3, S2, S3, S4, S5 (I'm assuming that Figs 2 and S1 are plotted directly from the raw data in Dryad?), as they are essential for readers to assess your analysis and to reproduce it. Please also ensure that figure legends in your manuscript include information on where the underlying data can be found.

REVIEWERS' COMMENTS:

Reviewer #1:

[identifies himself as Ruben Evens]

This Short Report represents the interesting discovery of lunar synchronization by Nightjars during their migration. The authors use a combination of modern tracking devices to state their novel findings. Although this Short Report presents exciting results, I have made a few general and specific comments.

General comments:

The introduction and results/discussion are well written (speculative at some points; which is not a bad thing!), but throughout the document it is rather difficult to appropriately allocate the findings to the methods used, or which data was actually used to come up with the results. Several sections were unclear and should/could be restructured or provided with additional information. For example, the authors state several times to have quantified foraging behaviour, but it was not described how this was done, or which criteria were applied (or I could not find them).

I hope the authors can forgive my remark that some of the results seem to be overstated. For example, although probably trough, on very few of the occasions 100% of the birds migrated in the same night. Indeed, we can see clear trends in synchronization of migratory movements, but perhaps this statement should/could be rephrased?

General comment to figures: please add info on full/new moon (white, black circle) and staging areas to all figures.

Flight activity data is not available in the Dryad folder (doi:10.5061/dryad.41vn01v).

Specific comments:

Lines 82-83: The authors say they distinguish two hypotheses, but in fact they attempt to find a plausible explanation for the first hypothesis. 

Line 94: The authors state that they observe reoccurring events of 100% of their tracked birds migrating simultaneously, however this only happened during autumn migration 2015 (n=4) and spring migration 2017 (eight birds for one night).

Line 115: Should this be “Fig. S2”?

Line 170: I find it very difficult to entangle how was discriminated between “daily foraging behaviour” and “migration activity”. Could this be explained a bit better? Furthermore, on line 79, how was this quantified?

Lines 212-214: What do the authors mean with this? Were these data stored during migration? Or were general data collected on standard, fixed positions?

Lines 212-228: I understand what the authors describe, but due to the long sentences, it is very hard to follow what they actually did. Would it be possible to simplify this section?

Line 360: Would it be possible to name the years differently? Each year tags were deployed actually comprises data of two subsequent years (autumn Y1 and spring Y2). This is very confusing.

Line 368: This figure is rather complex and not straight forward to interpret, I’m afraid. Could it be that the box plots complicate the interpretation of this graph? Do I understand correctly that you want to mention/hint on the correlation between the arrival date at stopovers and the expected duration of the stop over? Could this be depicted by actual observations and the regression line? Perhaps it is better to split this figure? Furthermore, in your text on line 121, your refer to a “fixed decision rule”. What do you mean with this, are some birds leaving in sub-optimal conditions? There seems to be quite some variation in the departure timing (except Temperate zone, Autumn), also mean departure timing is later than expected for the Sahel zone in Spring.

I have made all my comments objectively, and applaud the work that has been done to investigate the migration ecology of Nightjars.

Ruben Evens

Reviewer #2:

This study represents a very nice application of logging devices to capture in some detail the movement biology of a night-activity migratory bird. The largely speculative interpretation of the movement data is intriguing and nicely hypothesis driven viz. seeking explanations. Whereas it is not possible to draw firm conclusions about say fuel deposition rates, foraging behavior during stopover (not to mention during migratory flights), availability/abundance of prey in relation to foraging, the assumptions are clearly stated. Might be useful to the reader to be a bit more explicit about caveats -- what is not known and being assumed in this study not to mention what might be a future step test more rigorously the interpretation of the movement data (recognizing space limitations).

Reviewer #3:

[identifies himself as Jean-Michel Gaillard]

This work provides an orignal and timely analysis of the influence of lunar cycle on activity patterns of a nocturnal bird, the European nightjar. Based on an intensive study of 38 birds monitored for one year or less (i.e. for one or two migration events), the authors conclude that, by allowing birds to fuel their body reserves, the lunar cycle drives migration patterns of this nocturnal bird, with a high synchrony of migration movements about 10 days after a full moon event.

This manuscript potentially fits the requirements of a Short Report because the sample size is quite limited (only 38 birds monitored for only 1 year or less) but the findings are convincing and the reported work offers a significant and original contribution to improve our understanding of the key biological process of migration. Moreover, this manuscript is clearly written and the statistical analyses sound good to me. I did not find any major flaw in the present work. However, I found that a better consideration of individual heterogeneity, which nowadays is recognized to affect strongly movement patterns including migrations, is required. As the manuscript currently stands, there is only a brief mention that the bird identity was included as a random effect in the models (I guess as a random intercept but not as a random slope). By doing that, the authors satisfy to statistical requirements (i.e. including individual as a random intercept allows accounting for pseudo-replication issues) but totally neglect the biological significance of individual heterogeneity. A more thorough analysis (involving models with bird id included as a random slope at least for birds that were monitored for two migration events) and report (no information about the magnitude of the individual heterogeneity as measured by the random intercept is provided in the current version) are thus badly needed.

Detailed comments:

l. 31: « tactics » instead of « strategies »

l. 48: « tactics » instead of « strategies »

l. 60: « tactic » instead of « strategy »

l. 141: « tactic » instead of « strategy »

l. 161: The period indicated here (2015-2017) does not match with the individual monitoring periods provided in table S1!

J.M. Gaillard

---

## [Editor Report · Decision Letter 2]

11 Sep 2019

Dear Dr Norevik,

On behalf of my colleagues and the Academic Editor, Lars Chittka, I am pleased to inform you that we will be delighted to publish your Short Reports in PLOS Biology. 

PRESS 

Kind regards,

Hannah Harwood

Publication Assistant, 

PLOS Biology

on behalf of

Roland Roberts,

Senior Editor

PLOS Biology